# communications
# engineering

# Space object identification via polarimetric satellite laser ranging

Nils Bartels [1]✉, Paul Allenspacher[1], Daniel Hampf[1], Bernhard Heidenreich[2], Denise Keil[1], Ewan Schafer [1] &
Wolfgang Riede[1]

Low Earth orbits are becoming congested. The rapid identification and precise orbit determination of space objects is mandatory for space management. Satellite laser ranging (SLR) enables precise orbit determination by measuring the two-way photon travel time of laser pulses from a ground station to satellites equipped with retroreflectors. Here we propose polarization-modulated SLR, where specially designed retroreflectors positioned on a satellite switch the polarization state of received polarized photons and reflect them back to a ground station for analysis. Satellite identifiers can be coded into arrays of reflectors with different polarizing properties, while the orbit determining capability of conventional SLR is maintained. We design, build and test polarized light-switching retroreflector assemblies and investigate the feasibility of accurate signal measurement from SLR ground stations. The approach is passive, straightforward to integrate and requires no electricity. Polarization-modulated SLR could contribute to increasing demands of space object monitoring, for example of mega-constellations or during cluster launches.

[1] Deutsches Zentrum für Luft- und Raumfahrt (DLR), Institut für Technische Physik, Pfaffenwaldring 38-40, 70569 Stuttgart, Germany. [2] Deutsches Zentrum für Luft- und Raumfahrt (DLR), Institut für Bauweisen und Strukturtechnologie, Pfaffenwaldring 38-40, 70569 Stuttgart, Germany. ✉email: nils.bartels@dlr.de

Recent trends towards the miniaturization of satellites using standardized and affordable hardware components have led to a large increase in satellite launches and space traffic[1]. For example, the Indian Polar Satellite Launch Vehicle PSLV-C37—launched in 2017—deployed a record number of 104 satellites in sun-synchronous orbits[2]. This trend is greatly enhanced by new applications of space technology such as satellite communication and satellite internet, which require large satellite constellations (e.g., thousands of satellites in the Starlink constellation from SpaceX) to distribute these services worldwide.

While this development offers fascinating new technological and scientific opportunities, it also provides new challenges, since the increased traffic in space increases the risk of collisions and needs to be closely monitored by space situational awareness (SSA)/space traffic management (STM) systems. Especially for CubeSats (miniaturized satellites made up of multiple cubic modules of $11.35 \text{ cm} \times 10 \text{ cm} \times 10 \text{ cm}$ size), which are often deployed in cluster launches within a short period of time, it can take weeks to months to identify the satellites and up to 20% may never be claimed[3–5]. This situation has recently been described as the "CubeSat confusion", meaning that satellite operators have difficulty making contact with their satellite and that the satellite will not be registered with governmental agencies, which is an obligation according to the outer space treaty and important to keep track of traffic in space[3]. On top of the increasing space traffic from satellites, there is also an increasing density of space debris objects[1] e.g., due to explosions of spacecraft from left-over energy (fuel and batteries), collisions between satellites and/or existing space debris, or the usage of anti-satellite weapons (the latest test of an anti-satellite weapon was conducted during the revision process of this manuscript and destroyed the satellite Kosmos 1408).

As a contribution towards overcoming this confusion in space, it has been suggested to equip satellites with an identification beacon that can be read from ground[6]. The current status is that there is no widely used or accepted solution for such a beacon. The difficulty is that the beacon should ideally fulfill many requirements, in particular, a simple integration into different types of satellites, a high reliability and long lifetime, and a low mass and power consumption. Thus, different technologies for such an identification beacon are under development, see Fig. 1.

One approach is to equip satellites with an optical emitter producing coded light signals that can be detected on Earth with an optical telescope and single photon detectors (see Fig. 1a). Examples for this approach are the satellite LEDSat[7] and the extremely low resource identifier (ELROI)[6]. A second approach is the identification of a satellite via a radio transmitter (Fig. 1b). A low resource radio transponder is being developed by SRI International under the name CubeSat Identification Tag (CUBIT)[8]. In order to save electrical power and to avoid radio interference, this device will only send a radio signal once it is interrogated by a ground station.

A third potential approach is satellite laser ranging (SLR), see Fig. 1c. In SLR, a pulsed laser is directed towards the satellite and is retroreflected towards the ground station by one or more retroreflectors. These retroreflectors typically have three perpendicular reflecting surfaces and are thus called corner-cube retroreflectors (CCRs). The reflected light is then received by a telescope and photon detectors. The distance from the ground-station to the satellite can be calculated from the photon travel time, which is typically measured via an event timer. Advancements in SLR technology have allowed for great improvements in the ranging accuracy[9,10]. Thus, it is often possible to infer the number of retroreflectors placed on a satellite, unless the satellite is tilted directly towards the ground station so that the retroreflectors have the same range[11–13]. This means that a certain

number of satellites can be distinguished by mounting a different number of CCRs, but this approach is of course limited by the number of CCRs that can be attached.

Finally, a fourth approach is modulated retroreflectors (Fig. 1d). Here, the intensity of the retroreflected light is modulated with an electro-optical shutter, e.g., via multiple quantum well or liquid crystal modulators[14–16]. These have been suggested to be used on satellites for laser communication and could also provide status information and an ID of the satellite[17,18].

In this work, we suggest a method on how the number of distinguishable satellites via SLR (as described in Fig. 1c) can be greatly increased without using electrical components on the satellite by performing polarimetric SLR measurements. This concept benefits from recent advancements in SLR towards high repetition rate[19–21], which allows for a fast detection of the signal strength for the emission and detection of photons with different states of polarization. This requires that the satellite is equipped with one or more passive-optical assemblies that consist of a retroreflector and additional polarization optics that are mounted to the front face of the retroreflector. The change of polarization induced by these assemblies can be retrieved from ground by performing polarization-modulated SLR measurements. The retroreflector assemblies and the data evaluation in the polarimetric SLR experiment are designed in such a way, that the assemblies can be identified independently of their spatial orientation relative to the SLR ground station. Thus, these assemblies can be combined on the satellite to generate a large number of distinguishable satellites. Since SLR also offers the opportunity of measuring precise orbits, this technology offers a way to keep track of the identity and orbits of a large number of satellites or other space objects.

## Results

### Concept for the polarimetric identification of space objects.
The proposed setup for the polarimetric identification of a satellite via SLR is depicted in Fig. 2.

Similar to existing SLR ground stations, ranging is performed with a pulsed laser. The range resolution of the entire SLR station needs to be sufficient to resolve the signal coming from different retroreflector assemblies mounted to the satellite. This requires short laser pulses (picoseconds) and a high temporal resolution of the photon detector (typically a single-photon avalanche diode) and event timer used for counting and measuring the travel time of photons, which are reflected from the retroreflectors and collected with a telescope.

As opposed to conventional SLR, the ground station is additionally equipped with a polarization state generator (PSG) mounted to the exit of the laser system and a polarization state analyzer (PSA) mounted between the telescope and the photon detector. The PSG can be switched on a short time scale (a few milliseconds) to generate a modulated emission of laser pulses with either right circular (RC) or left circular (LC) polarization. This fast switching can for example be achieved via liquid crystal variable retarders, which have already been used to perform polarimetric LIDAR measurements[22]. A delay generator can be used to trigger the controllers of the liquid crystal variable retarders with the appropriate time delays. The modulated laser beam is then directed towards the satellite, which is equipped with one or more retroreflector assemblies that retroreflect the beam towards the SLR ground station. For the retroreflector assemblies, we suggest to use a combination of a metal-coated retroreflector and additional polarization optics. In particular, we have analyzed a design with an outer quarter waveplate and a central polarization optics that can be either a second quarter waveplate or a (wiregrid) polarizer. Via the choice of the central polarization optics (quarter

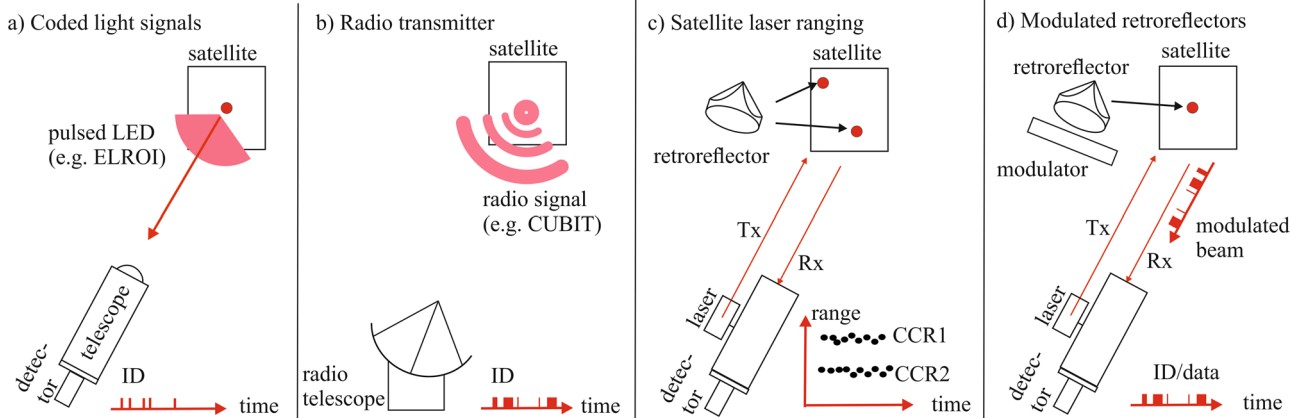

**Fig. 1 Identification concepts for satellites. a** Coded light signals (e.g., from a light-emitting diode, LED) are emitted at the satellite and detected on Earth with an optical telescope. An example for this technology is the Extremely Low Resource Optical Identifier (ELROI). **b** A radio signal carrying an ID is detected by a radio telescope. An example for this technology is the CubeSat Identification Tag (CUBIT). **c** In satellite laser ranging (SLR) emitted laser light (Tx) is retroreflected from one or more retroreflectors (red spots on the satellite) and the signal Rx (the red dots represent detected photons) is detected at the SLR ground station. **d** Modulated retroreflectors use electro-optical shutters to modulate the retroreflected intensity of a laser beam. This can be used for laser communication or provide an ID of the satellite.

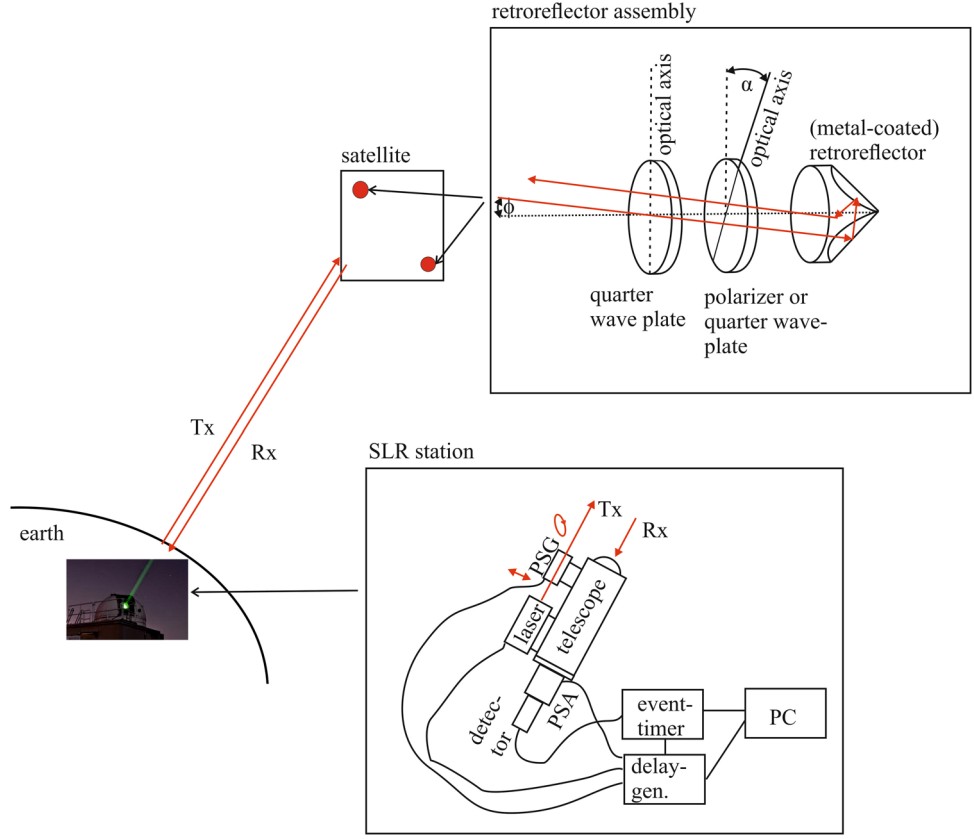

**Fig. 2 Schematic setup for polarization-modulated SLR.** A satellite laser ranging (SLR) ground station emits polarized laser radiation (Tx) towards the satellite. The polarization is set with the polarization state generator (PSG). Retroreflector assemblies mounted to the satellite alter the polarization state of the retroreflected signal (Rx). These photons are gathered by the telescope, pass the polarization state analyzer (PSA), and are detected with a photon detector.

waveplate versus polarizer) and by selecting a specific rotation angle $\alpha$ between the optical axes of the two polarization optics, it is possible to generate assemblies that can be uniquely identified from ground via polarimetric SLR. At the SLR ground station, the retroreflected photons are gathered with a telescope and directed towards the PSA and the photon detector. Similar to the PSG, the

PSA can be switched to transmit RC or LC components of the received signal. By using an event timer, each photon will be correlated with the photon travel time as well as the polarization state (RC versus LC) set at the PSG and the PSA.

Figure 3 shows the suggested detection principle for the polarimeteric identification of a satellite.

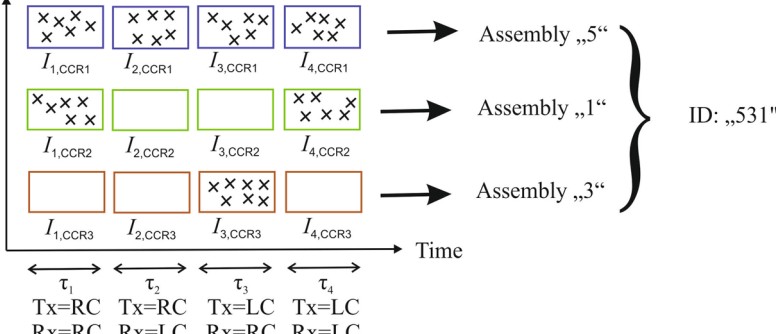

**Fig. 3 Detection principle.** Detection principle for the polarimetric identification of a space object. During the periods $\tau_1$ to $\tau_4$, the emitter (Tx) and the receiver (Rx) are switched between right-circular (RC) and left-circular (LC) polarization. The detected photons are shown as crosses. The information of the relative intensities (number of crosses) for a specific corner-cube retroreflector (CCR) assembly (or in other words a specific photon travel time) for each $\tau$ is used to identify the satellite. In this example, the space object is equipped with 3 retroreflector assemblies and yields the ID "531".

In the shown example, the satellite is equipped with three retroreflector assemblies. At the SLR ground station, the polarization of the transmitted laser (Tx) generated in the PSG and of the detected signal (Rx) are periodically modulated during the periods $\tau_1$ to $\tau_4$. This modulation is repeated on a fast timescale (e.g., 50 ms per time interval) in order to separate the intensity modulation from other fluctuations in the signal intensity (e.g., atmospheric turbulence, tracking errors, or fluctuations in the laser intensity). With this design, each detected photon will be correlated with the polarization period ($\tau_1$ to $\tau_4$) as well as with a specific retroreflector assembly via its photon travel time. Avoiding a bias of these intensities by noise (meaning photo-electrons introduced by background light or intrinsic detector noise), may require using techniques for noise reduction such as using spectral filters (that block sunlight, but transmit the SLR laser wavelength) in front of the photon detector, to limit the field-of-view of the receive telescope[23] and/or to apply filtering algorithms to separate signal from noise in the data evaluation[24].

Thus, for each retroreflector assembly four intensities will be measured

$$
\begin{aligned}
I_1 &= I(T_x = RC, R_x = RC) \\
I_2 &= I(T_x = RC, R_x = LC) \\
I_3 &= I(T_x = LC, R_x = RC) \\
I_4 &= I(T_x = LC, R_x = LC).
\end{aligned}
\tag{1}
$$

Using the definitions $I_a = I_1 + I_2$ and $I_b = I_3 + I_4$, we introduce the following symmetry parameters

$$
P_1 = \frac{I_a - I_b}{I_a + I_b}, \quad P_2 = \frac{I_1 - I_2}{I_1 + I_2}, \quad P_3 = \frac{I_3 - I_4}{I_3 + I_4}
\tag{2}
$$

which can take values in the range from +1 to −1.

Table 1 shows results for the calculation of the symmetry parameters of seven different retroreflector assemblies. The calculation uses the Mueller matrix algorithm[25] and is described in Supplementary Note 1. The assemblies either use a true zero-order quarter waveplate (assemblies 1–2) or a polarizer (assemblies 3–7) as the central polarization optics. Furthermore, the assemblies have a different angle $\alpha$ between the optical axes of the polarization optics. The angle $\alpha$ is selected to maximize the difference in the intensities and symmetry parameters, to allow for a robust identification of the satellite. It should however be noted, that assemblies no. 3 and 4 (the same is true for assemblies no. 6 and 7) have the same positive/negative sign and only a 3 dB difference between their intensities. These different intensities

have to be resolved in the polarimetric SLR experiment, which will be influenced by effects such as atmospheric turbulence and pointing errors. An option to unburden this high demand from the SLR experiment would be to decide that assemblies 4 and 6 will not be used. In that case, there would only be 5 distinguishable CCR assemblies.

A key feature of these assemblies is that the symmetry parameters are (at least to a first approximation) independent of the incidence angles $\phi$ and $\theta_i$ of light entering the retroreflector assemblies, which is why they can be used to identify the satellite.

The "identification number" of the satellite is then the combination of the assembly numbers of each assembly (e.g., "531" in Fig. 3). According to combinatorics, the number of combinations with repetition (meaning that each assembly can be used more than once) and without considering the sequence for n different assembly types and k mounted assemblies (where the SLR signal can be resolved in time) is given by the binomial coefficient $\binom{n+k-1}{k}$. As an example, for $k = 3$ mounted assemblies and $n = 7$ assembly types (as given in Table 1) this would give a number of 84 different IDs, which would be sufficient to provide a unique ID for all satellites of most cluster launches. If CCR assemblies 4 and 6 are omitted, there would be 35 different IDs for $k = 3$ mounted assemblies. In addition to increasing the number of mounted CCRs, a way to increase the number of IDs could be to expand the concept towards multi-wavelength SLR[13,26,27] in combination with color filters. Another approach could be to not only detect the LC and RC components, but rather the full Stokes vector of the retroreflected light. In this case, the already presented assemblies would span the full Poincaré sphere in the received polarization. Of course, such an approach is experimentally challenging, because the full analysis of the Stokes vector at the receiver requires measuring 6 intensities (with RC, LC, horizontal, vertical, diagonal, and anti-diagonal polarization).

It is clear, that the number of IDs that can be distinguished via polarimetric SLR is much smaller than the number of satellites used in mega-constellations or the number of CubeSats in space. However, polarization-modulated SLR can still contribute in specific ways to SSA/STM systems such as the US Space Surveillance Network (US-SSN) or the newly established European Union Space Surveillance and Tracking network. These networks use a combination of different technologies (e.g., tracking radar, detection radar, imaging radar, optical telescopes) to constantly monitor objects in space and to keep "track custody". For the identification of a satellite, the first critical

**Table 1 Calculated symmetry parameters for different retroreflector assemblies.**

| Assembly | Design[a] | $\alpha$ | $P_1$ | $P_2$ | $P_3$ | Intensities |
|---|---|---|---|---|---|---|
| 1[b] | A | 0° | 0 | 1 | −1 | $(I_1 = I_4) > 0$; $I_2 = I_3 = 0$ |
| 2 | A | 45° | 0 | −1 | 1 | $I_1 = I_4 = 0$; $(I_2 = I_3) > 0$ |
| 3 | B | −45° | −1 | $\lim\limits_{\alpha \to -45°} P_2(\alpha) = 1$ | 1 | $I_1 = I_4 = I_2 = 0$; $I_3 > 0$ |
| 4 | B | −15° | −0.5 | 0.5 | 0.5 | $I_3 > (I_1 = I_4) > I_2$ |
| 5 | B | 0° | 0 | 0 | 0 | $I_1 = I_2 = I_3 = I_4$ |
| 6 | B | 15° | 0.5 | −0.5 | −0.5 | $I_2 > (I_1 = I_4) > I_3$ |
| 7 | B | 45° | 1 | −1 | $\lim\limits_{\alpha \to 45°} P_2(\alpha) = -1$ | $I_1 = I_4 = I_3 = 0$; $I_2 > 0$ |

[a]Design A uses two quarter wave plates whereas design B uses a quarter wave plate and a polarizer mounted to the front face of the retroreflector.
[b]Assembly 1 has the same symmetry parameters as a metal-coated retroreflector without additional polarization optics.

moment is when satellites are deployed into space. SSA/STM systems have difficulties with the initial identification, if several satellites with identical shape (especially CubeSats) are launched into very similar orbits within a short period of time, such that these systems have insufficient time to follow along[3]. Thus, if satellites launched within a short time have a different polarimetric ID, all launched satellites can be initially identified. A re-identification of a satellite would then only be necessary, if the track of a satellite is "lost", because there are interruptions in the observation with the SSA/STM network in combination with track changes (e.g., due to an unexpected maneuver or sudden increases in solar drag) or when two satellite orbits become confusingly close[6]. Of course, in these situations it would be desirable if all satellites had a different ID, but in most cases, a limited number of IDs would be enough to associate a track back to the corresponding satellite. For example, the situation when two satellite orbits become confusingly close would only be a problem, if both satellites had the same shape (resolvable with a tracking Radar), the same polarimetric ID and if no communication (typically via radio commands) to the satellites is possible (e.g., because of satellite outages). The likelihood of such events can be greatly reduced by distributing the IDs wisely, such that satellites with similar orbits have different IDs.

**Laboratory testing of the polarization properties of retroreflector assemblies.** We designed and built an experimental setup in order to test the polarimetric properties of the suggested retroreflector assemblies (see Fig. 4 and the Methods section for experimental details). This setup allows for an automatic data acquisition of the polarization-dependent intensities $I_1$ to $I_4$ from which the symmetry parameters $P_1$ to $P_3$ can be calculated.

Figure 5 shows a comparison between the calculated and measured symmetry parameters for all 7 retroreflector assemblies (as defined in Table 1). The experimental data points are given for normal incidence and the error bars describe the dependence of the symmetry parameters on the incidence angles $\theta_i$ and $\phi$. To derive these error bars, we have systematically measured the symmetry parameters over a wide range of incidence angles $\theta_i$ and $\phi$. For the interested reader this detailed data and its analysis are provided in Supplementary Note 2. If we limit the incidence angle to $\phi = \pm 30°$ for the assemblies 1 and 2 and to $\phi = \pm 40°$ for the assemblies 5–7, we find that symmetry parameters of the different assemblies are well separated independently of the incidence angles and can thus be used to identify the satellite.

The data provided in Fig. 5 was obtained by measuring the total (polarization dependent) power of light reflected from the retroreflector assembly. In an actual SLR experiment, the SLR ground station will not be able to measure the total power. Due to the large distance between the SLR station and the satellite (e.g., 600 km for a typical satellite in a low Earth orbit), the retroreflector will instead act as an aperture and the SLR ground station will only detect a small section of its far-field diffraction pattern (FFDP). Furthermore, the SLR station is not placed in the center of the diffraction pattern, due to a relativistic effect known as the velocity aberration[9,28]. Instead, the apparent position of the SLR station in the FFDP will typically be at scattering angles ($\Theta_x$ and $\Theta_y$) between 21 and 53 µrad for a satellite in a low Earth orbit[29]. This means, that the symmetry parameters should ideally be homogeneous within this range of the diffraction pattern. In our experimental setup (Fig. 4) we have thus included the opportunity to measure the polarization-dependent FFDP of our retroreflector assemblies. It should be noted, that the retroreflector assemblies are tested in air. This represents a simplification, since the assemblies are designed to be placed in space (vacuum). This can affect the FFDP, especially in situations where the retroreflector is irradiated by sunlight[30].

As an example result, Fig. 6 shows the FFDP of CCR assembly no. 6 at incidence angles of $\phi = 30°$ and $\theta_i = 0°$ as measured for four different combinations of polarization states corresponding to the intensities $I_1$ to $I_4$. We find that the diffraction patterns measured for different combinations of emitted and detected polarization states have a very similar shape although the intensities are very different (e.g., up to 3600 units in panel B and only up to 360 units in panel C of Fig. 6).

To further analyze this data, we calculated the symmetry parameters $P_1$ to $P_3$ at each position of the FFDP, see Fig. 7. Ideally, the analyzed assembly should have the symmetry parameters $P_1 = 0.5$ and $P_2 = P_3 = -0.5$. We find that the symmetry parameters have a variation of less than ±0.1 (additional data is provided in Supplementary Note 3) within the relevant range of the diffraction pattern (between diffraction angles of 21 and 53 µrad as indicated by dashed circles in Fig. 7). In addition, the polarimetric SLR signal returning from a satellite passing the SLR station will have different scattering angles during the overflight, leading to an averaging of errors introduced by the polarization dependence of the FFDP. We thus conclude that it is feasible to measure the symmetry parameters with high accuracy, even though the SLR station will only measure a small section of the diffracted beam.

The measured FFDPs presented in this work represent the signal originating from a single retroreflector. In a situation where the light travel time for multiple CCRs overlap temporally, the FFDP of two retroreflectors will experience an intensity modulation due to interference similar to a double-slit experiment. This interference can however not be observed on Earth, because the phase difference of the light entering the two CCRs mounted to the moving satellite will randomly fluctuate on a millisecond timescale and thus average out to the FFDP of a single CCR. Note, that the measurement scheme is anyway unaffected from this additional interference as it relies on the temporal separation between pulses, for which interference is avoided.

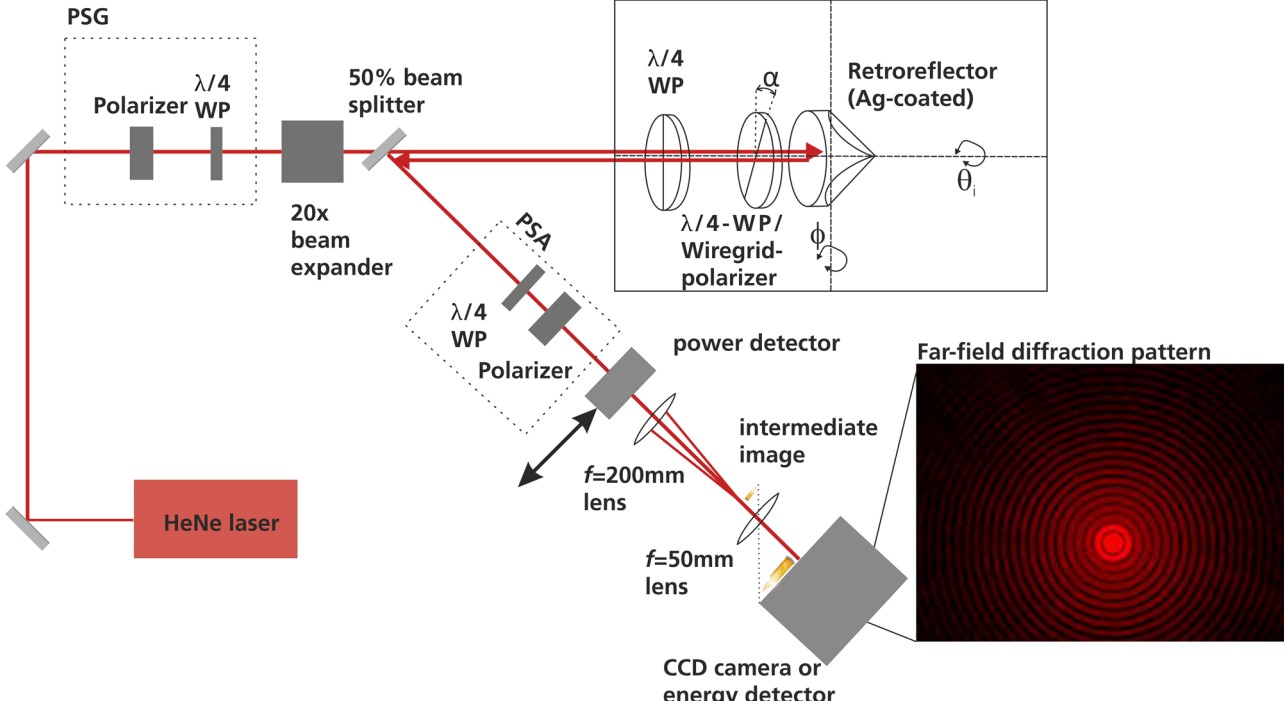

**Fig. 4 Laboratory test setup.** Laboratory test setup for measuring the polarization properties of different retroreflector assemblies. Further details of the setup are explained in the "Methods" section.

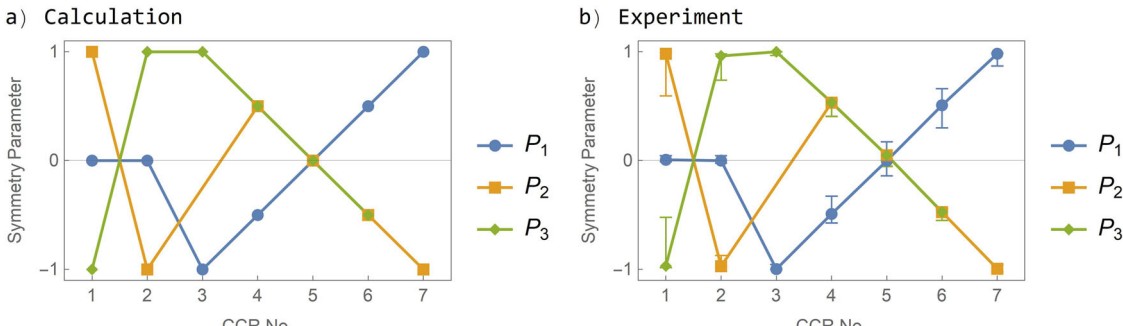

**Fig. 5 Symmetry parameters of retroreflector assemblies. a** Calculated symmetry parameters for different corner-cube retroreflector (CCR) assemblies as defined in Table 1. The calculation is described in Supplmentary Note 1. **b** Symmetry parameters measured in laboratory experiments. The error bars for the experimental data account for the dependence on the incidence angles and hold for of $\phi = \pm 30°$ for CCR assemblies 1 and 2 and for $\phi = \pm 40°$ for CCR assemblies 3–7.

**Challenges and chances for the polarimetric space object identification**. In order to assess technological challenges for the polarimetric space object identification, we gathered requirements for the retroreflector assembly and the SLR station (we currently work on upgrading our SLR station miniSLR[31] to allow for polarimetric SLR experiments) that will have to be met in order to achieve this goal.

Among these requirements, we consider the following points as particularly important:

- For a successful identification and a precise orbit determination, the satellite needs to pass over the SLR ground station with good observation weather. In order to obtain a timely identification after launch, it will thus be beneficial to use a network of several polarimetric SLR stations. Ideally, the SLR ground stations should be capable of daylight tracking[23] to maximize the potential measurement time of each SLR station. It is also required to have a

coarse knowledge of the satellite track (e.g., via TLE data provided by the US-SSN).

- If the ID of the satellite is given by more than one CCR (as shown in Fig. 3), the signal from the CCRs has to be resolved in time. The required range resolution depends on the distance between the mounting positions of the retroreflector assemblies on the satellite and is calculated in Supplementary Note 5. While the range resolution can be obtained even for a CubeSat, it is important that this requirement is fulfilled by the polarimetric SLR station.
- The SLR station and the assembly have to be optimized to achieve a high accuracy of the determination of symmetry parameters and thus a high confidence in the determination of the satellite's ID. This requires a careful testing and calibration of the polarization properties of the SLR station. Furthermore, the symmetry parameters and polarization-dependent FFDPs of the retroreflector assemblies need to be measured and documented over the

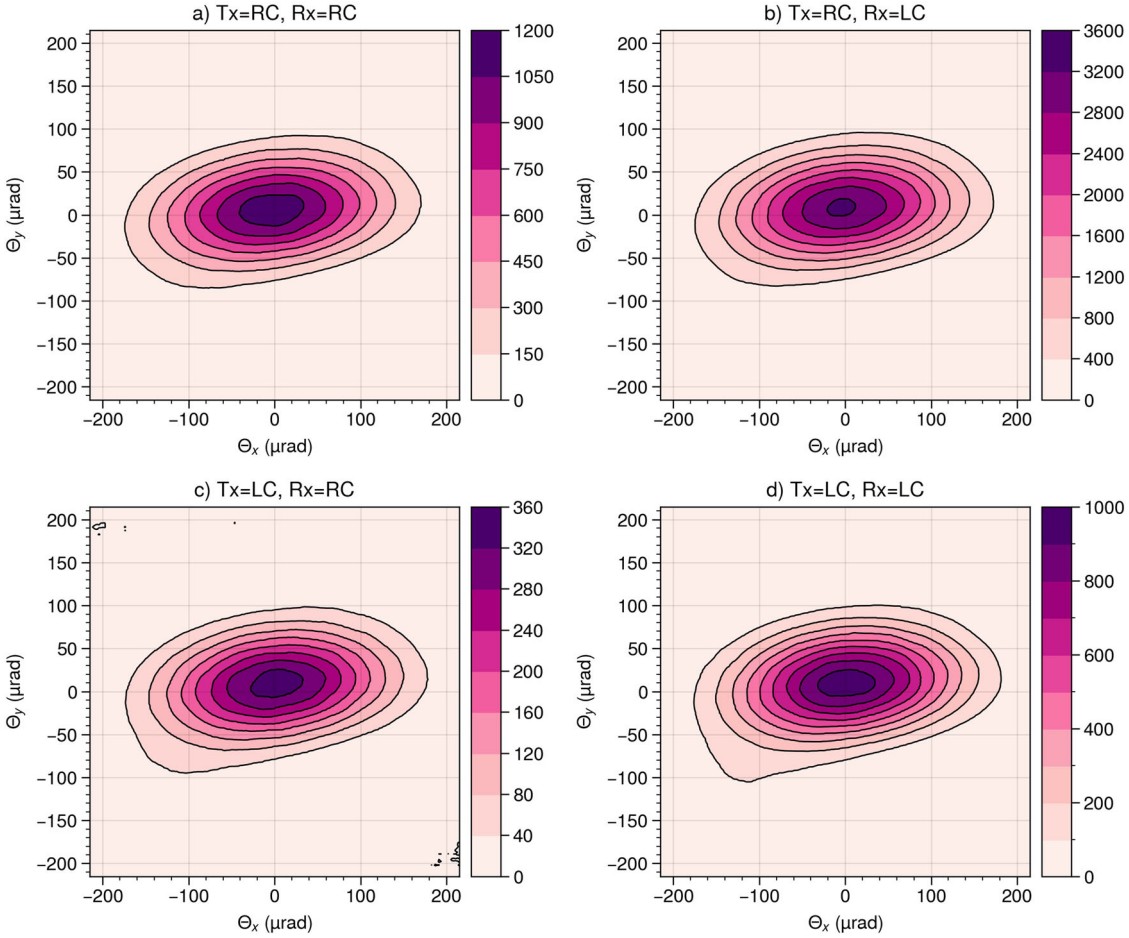

**Fig. 6 Polarization-dependent FFDP of retroreflector assembly 6. a** Far-field diffraction pattern (FFDP) measured at an incidence angle of $\phi = 30°$ when irradiated with right-circular polarization (Tx=RC) and detected with right-circular polarization (Rx=RC). **b** Same as panel a, but for Tx=RC and Rx=LC. **c** Same as panel **a**, but for Tx=LC and Rx=RC. **d** Same as panel **a**, but for Tx=LC and Rx=LC.

relevant range of incidence angles before sending these into orbit. Also, environmental effects such as solar radiation that might be incident to the CCR and lead to fluctuations in the diffraction pattern[30] or changes of polarization due to the atmosphere (which have fortunately been found to be below the detection limit in previous studies[32,33]) have to be taken into account. Additionally, it is important to achieve a sufficient photon count during the SLR experiment. As discussed in Supplementary Note 4, the detection of 1000 photo electrons would reduce the statistical error in the determination of symmetry parameters to less than ±0.05, which can be easily achieved with modern SLR stations for satellites in low Earth orbits.

- The retroreflector assemblies (in particular the polarization optics) need to be qualified for space usage.

Despite these challenging requirements, we see several potential advantages of this approach compared to other technologies for the identification of satellites (as introduced in the Introduction).

- The retroreflector assembly is completely passive optical, meaning that it does not require any electrical components on the satellite. This is beneficial, since electric circuits are prone to malfunctioning, especially under space conditions (space radiation). The passive optical design greatly simplifies the integration for different satellites and it can

be expected that the CCR assemblies will maintain their functionality over a long time. Retroreflectors are known to work for a long time, even after the end of life of a satellite mission. As an example, the satellite Beacon-C (NORAD ID: 1328) has been tracked by SLR since 1965.

- As an additional advantage, the SLR technology is laser-based and thus directed into a small solid angle (limited only by diffraction) compared to an undirected emission (e.g., from an LED). This greatly enhances the maximum range distance, which is impressively demonstrated by lunar laser ranging.

- Polarimetric SLR will not only provide IDs of satellites, but also highly accurate orbits. This will help to improve predictions for orbital data and thus help avoiding unnecessary maneuvers for collision avoidance.

In addition to these major advantages, by using specific mountings positions of retroreflectors (e.g., of three retro-reflectors) polarimetric SLR also yields attitude information about the satellite[34]. This attitude information can be used for an even further improvement of the orbit determination, because an attitude-dependent, center-of-mass range correction[35] can be applied. Additionally, if periodic oscillation is observed in the measured range residuals, this could imply tumbling motion, which could be used to diagnose problems with the satellite. We would like to further note, that a typical CCR used for satellites in low Earth orbit with a half inch (12.4 mm) diameter front face has a weight of 1.5 g only. This means that several mounted CCRs

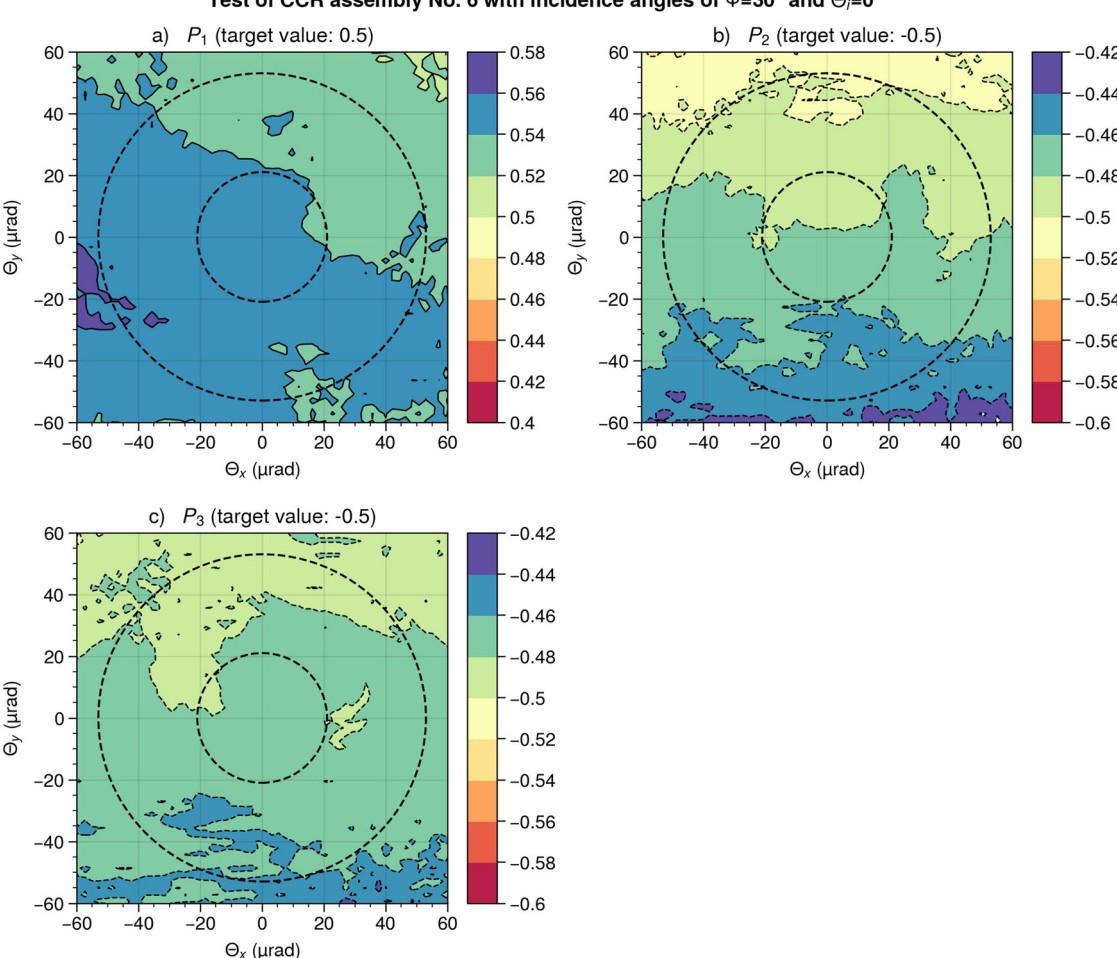

**Fig. 7 Symmetry parameters of CCR assembly 6. a** Symmetry parameter $P_1$ of corner-cube retroreflector (CCR) assembly 6 as a function of the diffraction angles $\Theta_x$ and $\Theta_y$ at an incidence angle of $\phi = 30°$. The dashed circles indicate the minimum (inner circle) and the maximum (outer circle) velocity aberration. **b** Same as panel **a**, but the symmetry parameter $P_2$. **c** Same as panel **a**, but the symmetry parameter $P_3$.

with additional polarization optics can easily be attached to CubeSats. In the long run, even lighter systems than retroflectors could be based on optical metamaterials[36] or photonic crystals with designed polarimatric properties.

## Discussion

We suggest a method for simultaneously determining precise orbits and for identifying space objects. This method uses polarimetric SLR of retroreflector assemblies. Major advantages of this approach are that the retroreflector assembly is completely passive, which allows for a very simple integration into the satellite. Since the assembly does not require electricity, it will furthermore work in case of a satellite outage. Additionally, the directed photon transfer (only limited by diffraction) is very favorable for the photon budget compared to an undirected emission of photons (e.g., via LEDs emitting in a large solid angular space). This technology is currently being further explored in the laboratory and we hope to prove its practical application in an upcoming space mission. It is our hope that this technology will be taken into consideration by satellite operators around the world and will thus help to address the "CubeSat confusion" in low Earth orbit and to reduce the number of collisions and unnecessary collision avoidance maneuvers.

## Methods

**Laboratory test setup**. As shown in Fig. 4. Briefly, the output of a continuous-wave Helium-Neon laser (Melles Griot 25-LHP-151-230, 632 nm) passes through a PSG that consists of a Glan-Thompson calcite polarizer (Thorlabs, GTH 10M) and a zero-order quarter waveplate (Thorlabs, WPQ10M-633). The beam is then expanded by a factor of 20 (to achieve a homogeneous illumination of the retro-reflector) and passes a 50% reflectivity, non-polarizing beam splitter before entering the retroreflector assembly. The retroreflector assembly consists of an outer true zero-order quarter waveplate (Thorlabs, WPQ10ME-633), a second polarization optics (either another true zero-order quarter waveplate or a polar-izer), and the retroreflector. The polarizer is a wire-grid polarizer (Thorlabs, WP25M-VIS), since these are much smaller than other types of polarizers. The retroreflector (Edmund Optics GmbH, 45-202) is made from n-BK7, has a dia-meter of 12.7 mm and the back facets have a protected silver coating. Both polarization optics and the retroreflector itself are mounted in rotational mounts. This way the incidence rotational angle $\theta_i$ (rotational angle between the trans-missive axis of the polarizer in the PSA and the fast axis of the outer quarter waveplate of the retroreflector assembly where the fast axis of this quarter wave-plate is coaligned with one edge of the back facets of the CCR) and $\alpha$ (angle between the polarization axes of the polarization optics within the CCR assembly) can be changed. Furthermore, all optics can be rotated by the incidence angle $\phi$, which is the angle between the laser beam propagation and the front facet of the CCR assembly. The retroreflected light then encounters the beam splitter for a second time, and the reflected light passes the polarization state analyzer (the PSA uses the same optical components as the PSG). Behind the PSA, the power of the reflected light can be measured with a power detector (Ophir, PD300-SH) to obtain an integrated intensity. A LabVIEW program was developed that reads the power from the power detector. The software also controls the orientation of the quarter-waveplates in the PSG and the PSA, which can be rotated via motorized rotational mounts.

**Measurement of far-field diffraction patterns**. For a measurement of the FFDP, we can remove the power detector from the experimental setup depicted in Fig. 4 and instead focus the reflected light with an $f = 200$ mm lens. The beam profile in the focal plane (which corresponds to the far-field image) is then imaged to a CCD camera (Ophir, SP928) with an $f = 50$ mm lens. The image scale (measured in pixels) is converted to scattering angles $\Theta_x$ and $\Theta_y$ via a calibration[37]. For this calibration, the retroreflector assembly is replaced by a circular aperture (with a known diameter $d$) placed in front of a silver mirror. We then detect the Airy diffraction pattern on the CCD camera. This diffraction pattern can be used for the calibration, since the first minimum of the diffraction pattern is known to occur at a scattering angle of $\theta = 1.22\lambda/d$.

## Data availability

All the data and methods are present in the main text and the supplementary materials. Any other relevant data are available from the authors upon reasonable request.

## Code availability

Code used for the calculation of symmetry parameters with the Mueller matrix algorithm (Supplementary note 1), for the calculation of the photon link budget (Supplementary note 4) and the estimation of the statistical error in the determination of symmetry parameters (also part of Supplementary Note 4) is provided as a notebook for the software Wolfram Mathematica available at https://figshare.com/articles/software/PolarimetricSLR_nb/16870984 or via the DOI as https://doi.org/10.6084/m9.figshare.16870984.v3.

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

## Acknowledgements

The authors acknowledge the funding of this work within the Helmholtz Associations Space Research Program in the Space Systems Technology research area. Furthermore, interesting scientific discussions within the scientific community of the International Laser Ranging Service (ILRS) are greatly appreciated.

## Author contributions

All authors developed the idea and the approach for the polarimetric satellite laser ranging during joint meetings. W.R. and N.B. designed the research. N.B., P.A., and D.K. built the polarimetric setup. N.B. and P.A. conducted the polarimetric experiments. N.B. and D.K. analyzed the data. N.B. wrote the initial manuscript. All authors reviewed and contributed to the submitted version of the manuscript.

## Funding

## Competing interests

The authors declare no competing interests.
