## [Peer Review File · Communications Engineering]

This manuscript has been previously reviewed at another Nature Portfolio journal. This document only contains reviewer comments and rebuttal letters for versions considered at Communications Engineering.

REVIEWERS' COMMENTS:

Reviewer #1 (Remarks to the Author):

Comment 1:

The author says:

"Polarimetric SLR with the mini-SLR will not require burst mode pulse collision avoidance. The reason for this is that the emitter and the receiver of the mini-SLR have a distance of approximately 1 m. Thus, photons from back-reflections in the up-going laser beam are not detected by the receive telescope. We added a discussion of this topic to our manuscript. "

As the Rayleigh scattering scales as $(1 + \cos^2(\theta))/2$, with a distance between the transmitter and the receiver of 1 m already at 10 m distance from the telescope the back reflection is $(1 + \cos^2(\arcsin(1/10)))/2 = 0.995$, i.e. 99,5% of what we would expect with a monostatic system. The authors should specify if they are using other techniques to reduce the detection from backscattering (i.e. angular filtering via field of view reduction) or to provide a reference of a bistatic system working without burst mode with single photon detectors and repetition rate and energy per pulse close to the one proposed.

Comment 2:

The authors satisfactorily replied to my comments.

Comment 3:

Provided comment 1 is satisfactorily addressed, also this comment can be considered closed.

Reviewer #2 (Remarks to the Author):

Dear Authors,

This paper presents the space object identification with polarimetric Satellite Laser Ranging (SLR) and the method to resolve the combination of several Corner Cube Reflector (CCR) assemblies. The authors proposed the polarimetric SLR with passive optical devices, and this point is an incremental improvement from the past studies. The paper was written well and explained clearly for this version. However, the applicability of the small number of the supplied IDs is weak for future applications of many debris, mega-constellations and a plenty of Cubesats in space in the future. I think the value of this paper can be improved if the number of the supplied IDs can be increased.

Recommendation:

In the free-space laser communication community, the circular polarizations are usually used for the free-space propagation. This is because the optical axis of the linear polarization in the moving terminals can be fixed against the optical bench inside the laser communication terminals, in which a quarter wave plate (QWP) is used in front of the internal optics. And the combinations of a QWP, a polarizer (POL) and a CCR are usually used for the optical calibration and the optical isolation purposes for the optical transmission and reception. Therefore, the usage of these combinations is not new and well known but I think if you use one half wave plate (HWP), you can improve the system. According to the knowledge, I suggest one improvement on your system.

If you insert one HWP between the POL and the CCR in your system, one freedom parameter could be

implemented with still only passive optical devices. This could increase the number of the supplied IDs in your system. If you consider one circular polarization case like the RHCP from the SLR station side, the polarization after the QWP will become a linear horizontal (or vertical) polarization in the CCR assembly. Then, you will additionally set a HWP with an angle, Beta [deg], between the input linear polarization axis and the fast axis of the HWP. After passing and reflected through the CCR, the polarization axis will experience the 4 times larger rotation angle, $(4 \times \text{Beta})$, and the intensity of the returned signal will be reduced by a factor of the coefficient of $\sin(4 \times \text{Beta})$ after the polarizer, for example.

In the different polarization case of the LHCP from the SLR station side, the linear polarization will become the orthogonal vertical (or horizontal) polarization contrary after the QWP. The procedure will be similar to that in the RHCP case, but the intensity of the returned signal will be reduced by a factor of the coefficient of $\cos(4 \times \text{Beta})$ after the polarizer. This could add the additional freedom parameter in your system as "Beta" and the additional combinations of P1 to P3 to distinguish the CCR assemblies. This modulation might be able to modulate P1 to P3 parameters independently (I am not sure in detail but please prove it). This parameter could be used as the arbitrary amplitude modulation parameters for P1 to P3 within the range less than the statistical error in the determination of the symmetry parameters as shown in Supplementary Figure 12.

Depending on the determination of the symmetry parameters within the measurement error, the number of IDs will be able to be increased. I have not proven completely this principle in detail, but please examine this method and verify them. I would think the concept must be improved.

Typo:

- Supplementary Information, Page 11,

The authors should consider the seeing as the worst case. The sentence should be replaced

"For the mini-SLR, the laser beam divergence is $50 \mu\text{rad}$ and thus smaller than the turbulence at good seeing (typically between 5 and $10 \mu\text{rad}$ [16])."

by

"For the mini-SLR, the laser beam divergence is $50 \mu\text{rad}$ and thus smaller than the turbulence at worst case seeing (typically between 5 and $10 \mu\text{rad}$ [16]).".

Thank you very much.

Response to the reviewers: “Space object identification via polarimetric satellite laser ranging”

Authors: Nils Bartels, Paul Allenspacher, Daniel Hampf, Bernhard Heidenreich, Denise Keil, Ewan Schafer, and Wolfgang Riede

Date: January 30th, 2022

We would like to thank the reviewers for their valuable time to review our manuscript.

Similar to our previous response letter, all reviewers' comments and questions are typeset in italic font. Our responses and remarks are written in plain font. Modification to the manuscript are written in blue color.

Response to Reviewer #1:

Reviewer #1 (Remarks to the Author):

The answer is completely satisfactory, and I recommend the publication of the paper in the current version.

The authors would like to thank the Reviewer 1 for his detailed comments and suggestions from the previous revisions.

Response to Reviewer #2:

Reviewer #1 (Remarks to the Author):

Dear Authors,

This paper presents the space object identification with polarimetric Satellite Laser Ranging (SLR) and the method to resolve the combination of several Corner Cube Reflector (CCR) assemblies. The authors proposed the polarimetric SLR with passive optical devices, and this point is an incremental improvement from the past studies. The paper was written well and explained clearly for this version. However, the applicability of the small number of the supplied IDs is weak for future applications of many debris, mega-constellations and a plenty of Cubesats in space in the future. I think the value of this paper can be improved if the number of the supplied IDs can be increased.

Recommendation:

In the free-space laser communication community, the circular polarizations are usually used for the free-space propagation. This is because the optical axis of the linear polarization in the moving terminals can be fixed against the optical bench inside the laser communication terminals, in which a quarter wave plate (QWP) is used in front of the internal optics. And the combinations of a QWP, a polarizer (POL) and a CCR are usually used for the optical calibration and the optical isolation purposes for the optical transmission and reception. Therefore, the usage of these combinations is not new and well known but I think if you use one half wave plate (HWP), you can improve the system. According to the knowledge, I suggest one improvement on your system.

If you insert one HWP between the POL and the CCR in your system, one freedom parameter could be implemented with still only passive optical devices. This could increase the number of the supplied IDs in your system. If you consider one circular polarization case like the RHCP from the SLR station side, the polarization after the QWP will become a linear horizontal (or vertical) polarization in the CCR assembly. Then, you will additionally set a HWP with an angle, β [deg], between the input linear polarization axis and the fast axis of the HWP. After passing and reflected through the CCR, the polarization axis will experience the 4 times larger rotation angle, $(4 \times \beta)$, and the intensity of the returned signal will be reduced by a factor of the coefficient of $\sin(4 \times \beta)$ after the polarizer, for example.

In the different polarization case of the LHCP from the SLR station side, the linear polarization will become the orthogonal vertical (or horizontal) polarization contrary after the QWP. The procedure will be similar to that in the RHCP case, but the intensity of the returned signal will be reduced by a factor of the coefficient of $\cos(4 \times \beta)$ after the polarizer. This could add the additional freedom parameter in your system as "Beta" and the additional combinations of P1 to P3 to distinguish the CCR assemblies. This modulation might be able to modulate P1 to P3 parameters independently (I am not sure in detail but please prove it). This parameter could be used as the arbitrary amplitude modulation parameters for P1 to P3 within the range less than the statistical error in the determination of the symmetry parameters as shown in Supplementary Figure 12.

Depending on the determination of the symmetry parameters within the measurement error, the number of IDs will be able to be increased. I have not proven completely this principle in detail, but please examine this method and verify them. I would think the concept must be improved.

If we understand your comment ("If you insert one HWP between the POL and the CCR in your system") correctly, your suggestion would be to use a quarter waveplate (QWP), followed by a half waveplate (HWP, rotated by the angle α), a polarizer (rotated by the angle β) and then the CCR. We

have calculated the Mueller matrix for this system (see the following snapshot from Mathematica). Unfortunately, the calculated Mueller matrix of this system is independent of the angle β and there is no effect of the half waveplate on the polarization of the retroreflected light.

Calculation of the polarization properties of additional retroreflector assemblies

Calculation of analytic formulas for the symmetry parameters

In[1] MLinearRetarderId[δ_-, θ_-] :=

$$\begin{pmatrix} 1 & 0 & 0 & 0 \\ 0 & \cos[2\theta]^2 + \sin[2\theta]^2 \cos[\delta] & \sin[2\theta] \cos[2\theta] (1 - \cos[\delta]) & -\sin[2\theta] \sin[\delta] \\ 0 & \sin[2\theta] \cos[2\theta] (1 - \cos[\delta]) & \sin[2\theta]^2 + \cos[2\theta]^2 \cos[\delta] & \cos[2\theta] \sin[\delta] \\ 0 & \sin[2\theta] \sin[\delta] & -\cos[2\theta] \sin[\delta] & \cos[\delta] \end{pmatrix}$$

(* Eq. S1 *)

In[2] MLinearDiattenuator[θ_-] :=

$$1/2 \begin{pmatrix} 1 & \cos[2\theta] & \sin[2\theta] & 0 \\ \cos[2\theta] & \cos[2\theta]^2 & \sin[2\theta] \cos[2\theta] & 0 \\ \sin[2\theta] & \sin[2\theta] \cos[2\theta] & \sin[2\theta]^2 & 0 \\ 0 & 0 & 0 & 0 \end{pmatrix} \quad (* \text{Eq. S2} *)$$

In[3] MM = $\begin{pmatrix} 1 & 0 & 0 & 0 \\ 0 & 1 & 0 & 0 \\ 0 & 0 & -1 & 0 \\ 0 & 0 & 0 & -1 \end{pmatrix}$; (* Eq. S3 *)

Case 1: Lambda/4, Pol, Lambda/2 -> Result: No effect of Lambda/2 waveplate

In[4] MCCRTTest[$\theta_-, \alpha_-, \beta_-$] := MLinearRetarderId[$(90^\circ), -(\theta)$].MLinearDiattenuator[$-(\theta + \alpha)$].
MLinearRetarderId[$(180^\circ), -(\theta + \beta)$].MM.MLinearRetarderId[$(180^\circ), \theta + \beta$].
MLinearDiattenuator[$(\theta + \alpha)$].MLinearRetarderId[$(90^\circ), \theta$]

In[5] Simplify[MCCRTTest[θ, α, β]] // MatrixForm

$$\text{Out[5]MatrixForm} \begin{pmatrix} \frac{1}{2} & \frac{1}{2} \cos[2\alpha] \cos[2\theta] & \frac{1}{2} \cos[2\alpha] \sin[2\theta] & \cos[\alpha] \sin[\alpha] \\ \frac{1}{2} \cos[2\alpha] \cos[2\theta] & \frac{1}{2} \cos[2\alpha]^2 \cos[2\theta]^2 & \frac{1}{4} \cos[2\alpha]^2 \sin[4\theta] & \frac{1}{4} \cos[2\theta] \sin[4\alpha] \\ -\frac{1}{2} \cos[2\alpha] \sin[2\theta] & -\frac{1}{4} \cos[2\alpha]^2 \sin[4\theta] & -\frac{1}{2} \cos[2\alpha]^2 \sin[2\theta]^2 & -\frac{1}{4} \sin[4\alpha] \sin[2\theta] \\ \cos[\alpha] \sin[\alpha] & \frac{1}{4} \cos[2\theta] \sin[4\alpha] & \frac{1}{4} \sin[4\alpha] \sin[2\theta] & 2 \cos[\alpha]^2 \sin[\alpha]^2 \end{pmatrix}$$

In[6] MCCRTestb[θ_-, α_-] := MLinearRetarderId[$(90^\circ), -(\theta)$].MLinearDiattenuator[$-(\theta + \alpha)$].
MM.MLinearDiattenuator[$(\theta + \alpha)$].MLinearRetarderId[$(90^\circ), \theta$]

In[7] Simplify[MCCRTestb[θ, α] - MCCRTTest[θ, α, β]] // MatrixForm

$$\text{Out[7]MatrixForm} \begin{pmatrix} 0 & 0 & 0 & 0 \\ 0 & 0 & 0 & 0 \\ 0 & 0 & 0 & 0 \\ 0 & 0 & 0 & 0 \end{pmatrix}$$

An approach to increase the number of IDs could be to perform a full analysis of the polarization (Stokes vector) of the reflected light from the CCRs, instead of just detecting its RC and LC components.

For example, if we use the assembly with two quarter waveplates and irradiate with right-circular polarized light (Stokes vector $(1,0,0,1)^T$), we obtain a retroreflected Stokes vector that is a function of

α and θ , see next picture.

Case 5 : Lambda/4, Lambda/4

```

In[19]= MCCRTest5[ $\theta$ ,  $\alpha$ ,  $\beta$ ] := MLinearRetarderId[(90°), -( $\theta$ )] . MLinearRetarderId[(90°), -( $\theta$  +  $\alpha$ )] . MM.MLinearRetarderId[(90°), ( $\theta$  +  $\alpha$ )] . MLinearRetarderId[(90°), ( $\theta$ )]
In[21]= Simplify[MCCRTest5[ $\theta$ ,  $\alpha$ ,  $\beta$ ]] // MatrixForm
Out[21]MatrixForm= 
$$\begin{pmatrix} 1 & & & & & \\ \frac{1}{4} (2 + 2 \cos[4\alpha] + \cos[4(\alpha - \theta)] - 2 \cos[4\theta] + \cos[4(\alpha + \theta)]) & & & & & \\ \theta & \sin[2\alpha]^2 \sin[4\theta] & & & & \\ 0 & \cos[2\theta] \sin[4\alpha] & & & & \\ 0 & & \frac{1}{4} (-2 - 2 \cos[4\alpha] + \cos[4(\alpha - \theta)] - 2 \cos[4\theta] + \cos[4(\alpha + \theta)]) & & & \\ & & & \sin[4\alpha] \sin[2\theta] & & \\ & & & & \cos[2\theta] \sin[4\alpha] & \\ & & & & & \cos[4\alpha] \end{pmatrix}$$

In[20]= Simplify[MCCRTest5[ $\theta$ ,  $\alpha$ ,  $\beta$ ] . {1, 0, 0, 1}] // MatrixForm
Out[20]MatrixForm= 
$$\begin{pmatrix} 1 & & & \\ \cos[2\theta] \sin[4\alpha] & & & \\ -\sin[4\alpha] \sin[2\theta] & & & \\ -\cos[4\alpha] & & & \end{pmatrix}$$

```

As obvious from the components of this Stokes vector, the variation of θ and α would span the full Poincaré sphere. For a satellite with a single CCR assembly, this would mean that is possible to obtain α (internal rotation angle between the quarter waveplates) and the orientation angle θ of the satellite in space.

In case of more than one CCR assembly mounted to the same satellite, it is possible to increase the number of ID's by not only varying the angle α , but also by using an offset $\Delta\theta$ between the angles θ of different CCR assemblies. Such an approach is beyond the scope of this work and experimentally challenging, because the full analysis of the Stokes vector at the receiver requires measuring 6 intensities (with right-circular, left-circular, horizontal, vertical, diagonal and anti-diagonal polarization).

Nonetheless, we have included the following sentence to the publication:

Old version:

In addition to increasing the number of mounted CCRs, a way to increase the number of IDs could be to expand the concept towards multi-wavelength SLR [13, 26, 27] in combination with color filters.

New version:

In addition to increasing the number of mounted CCRs, a way to increase the number of IDs could be to expand the concept towards multi-wavelength SLR [13, 26, 27] in combination with color filters. Another approach could be to not only detect the left-circular (LC) and right-circular (RC) components, but rather the full Stokes vector of the retroreflected light. In this case, the already presented assemblies would span the full Poincaré sphere in the received polarization. Of course, such an approach is experimentally challenging, because the full analysis of the Stokes vector at the receiver requires measuring 6 intensities (with RC, LC, horizontal, vertical, diagonal and anti-diagonal polarization).

Typo:

- Supplementary Information, Page 11,

The authors should consider the seeing as the worst case. The sentence should be replaced "For the mini-SLR, the laser beam divergence is 50 μ rad and thus smaller than the turbulence at good seeing (typically between 5 and 10 μ rad [16])."

by

"For the mini-SLR, the laser beam divergence is 50 μ rad and thus smaller than the turbulence at worst

case seeing (typically between 5 and 10 μrad [16])."

Thanks for this suggestion. We have modified page 11 of the supplementary material.

Old version:

For the mini-SLR, the laser beam divergence is 50 μrad and thus smaller than the turbulence at good seeing (typically between 5 and 10 μrad [16]).

New version:

For the mini-SLR, the laser beam divergence is 50 μrad and thus smaller than the turbulence at worst case seeing (typically between 5 and 10 μrad [16]).

Thank you very much.

Thank you!